# Perceived Determinants of Children’s Inadequate Sleep Health. A Concept Mapping Study among Professionals

**DOI:** 10.3390/ijerph17197315

**Published:** 2020-10-07

**Authors:** Laura S. Belmon, Fay B. Brasser, Vincent Busch, Maartje M. van Stralen, Irene A. Harmsen, Mai J. M. Chinapaw

**Affiliations:** 1Department of Public and Occupational Health, Amsterdam Public Health Research Institute, Amsterdam UMC, Vrije Universiteit Amsterdam, 1081 BT Amsterdam, The Netherlands; f.brasser@amsterdamumc.nl (F.B.B.); m.chinapaw@amsterdamumc.nl (M.J.M.C.); 2Sarphati Amsterdam, Public Health Service (GGD), City of Amsterdam, Nieuwe Achtergracht 100, 1018 WT Amsterdam, The Netherlands; i.a.harmsen@isala.nl; 3Department of Health Sciences, Faculty of Science and Amsterdam Public Health Research Institute, Vrije Universiteit Amsterdam, 1081 HV Amsterdam, The Netherlands; maartje.van.stralen@vu.nl

**Keywords:** sleep, professionals, childhood, children, determinants, factors, concept mapping, child public health, child health care

## Abstract

An increasing number of children experience inadequate sleep, which negatively effects their health. To promote healthy sleep among children, it is essential to understand the underlying determinants. This online concept mapping study therefore explores potential determinants of children’s inadequate sleep as perceived by professionals with expertise in the sleep health of children aged 4–12 years. Participants (*n* = 27) were divided in three groups: (1) doctors (*n* = 9); (2) nurses (*n* = 11); (3) sleep experts (*n* = 7). Participants generated potential determinants (i.e., ideas) of children’s inadequate sleep. Subsequently, they sorted all ideas by relatedness and rated their importance. These data were analysed using multidimensional scaling and hierarchical cluster analysis. The results of all three groups were combined and validated by an additional group of professionals (*n* = 16). A large variety of perceived determinants were identified. The most important determinants perceived by all groups belonged to the categories psychosocial determinants (i.e., worrying, a change in daily life), daytime and evening activities (i.e., screen use before bedtime, stimulating game play before bedtime, inadequate amount of daytime physical activity), and pedagogical determinants (i.e., inconsistent sleep schedule, lack of a bedtime routine). These perspectives are valuable for future longitudinal studies on the determinants of children’s sleep and the development of future healthy sleep interventions.

## 1. Introduction

Inadequate sleep health (i.e., insufficient sleep quantity and quality) [1] is becoming more prevalent among children. Children increasingly report shorter sleep duration, longer time to fall asleep, and daytime sleepiness [2,3,4]. This may lead to lower emotion regulation [5,6], a shorter attention span [5,6,7], reduced school performance [6,8,9], and an increased risk of overweight and diabetes later in life [5,10,11,12]. This rising prevalence and its negative consequences indicate a need for effective interventions to promote healthy sleep among children.

Effective preventive healthy sleep interventions require identification of the most important and changeable determinants of inadequate sleep [13]. A systematic review on longitudinal studies found that determinants of children’s inadequate sleep duration included inadequate past sleep health and a higher amount of screen time [14]. Of these, screen time is the only determinant that can be changed with behavioural interventions. Another review, which also included cross-sectional studies and trial/intervention studies, found that changeable determinants of children’s healthy sleep included establishing bedtime routines, maintaining a regular sleep schedule, setting age-appropriate bedtimes, ensuring a positive atmosphere at home, falling asleep independently, and meeting children’s emotional needs during the day [15]. In addition, a participatory mixed-method study identified the determinants of inadequate sleep perceived by children (aged 9–12 years) and parents, and these included fear, affective state, stressful situations, discomfort, physical well-being, sleep environment, energy, screen behaviour, stimulating activities, physical activity, diet, sleep schedule, family sleep habits, and social norms [16].

Children and parents are important stakeholders with valuable knowledge about children’s sleep health and its determinants. Other stakeholder groups with potential valuable insights include professionals with practical and clinical experience with children’s sleep health. Such professionals can also provide estimates of the occurrence of these determinants in practice. This knowledge can inform future healthy sleep intervention development and give direction to future research on determinants of child sleep health.

Therefore, we aim to explore the perspectives of professionals with expertise in children’s sleep health on potential determinants of inadequate sleep health among children aged 4–12 years.

## 2. Materials and Methods

An online concept mapping study was conducted to identify professionals’ perspectives on the potential determinants of inadequate sleep health among children aged 4–12 years [17]. This age group was selected as this marks the period for primary school in The Netherlands. In a participatory and collaborative process, professionals generated ideas about potential determinants during an individual brainstorm session and, subsequently, rated these ideas on their potential influence on children’s sleep and their occurrence in practice via a concept mapping tool (Ariadne). Concept mapping is a six-step process (see Figure 1) [16,17,18]. The first five steps are described below. Step six includes the identification of perceived determinants that may be relevant for future healthy sleep interventions (see Discussion). Concept mapping has been previously used to explore perspectives on behavioural determinants [16,19,20]. Additional information about this method can be found elsewhere [17].

### 2.1. Step (1) Preparation

A focus statement and two rating statements were created. These are questions or statements that instruct participants to generate ideas and assess the importance of those ideas. The purpose of the focus statement is to elicit ideas about the topic of interest, whereas the rating statement provides comparative ratings of importance for the generated ideas. The focus statement was “Inadequate sleep among children aged 4 to 12 may be due to …”. The rating statements were (1) “How much does this [idea] negatively influence children’s sleep?”, answered on a five-point Likert scale from “no influence at all = 1” to “great influence = 5”; and (2) “How often does this [idea] occur in practice?”, answered on a five-point Likert scale from “never = 1” to “always = 5”. The comprehensiveness and feasibility of the concept mapping tasks were pre-tested with five professionals from the Public Health Service of Amsterdam and changes were made accordingly.

All professionals were recruited via email. The head of the Child Public Health department in Amsterdam sent an email to all child public health nurses, doctors, and parent-and-child advisors in Amsterdam, and a member of the Dutch Association of Paediatrics sent an email to all its members. The email included an invitation to participate in the study, an information letter about the study, and a consent form. Participants were recruited through purposive sampling to select participants who were proficient in Dutch and had knowledge of and experience with sleep health and child health care among children aged 4–12 years [21].

Three groups were formed and labelled as (1) Doctors (i.e., child public health doctors); (2) Nurses (i.e., child public health nurses and parent-and-child advisors); (3) Sleep experts (i.e., scientists and doctors specialised in children’s sleep).

Participants filled out a general survey to obtain demographic information. Data were collected in March 2018. Participation was rewarded with a ten euro online gift card. The Amsterdam UMC, Vrije Universiteit Amsterdam Ethical Committee approved the protocol (study protocol 2017.013). Participants gave informed consent by signing a digital consent form.

### 2.2. Step (2) Generating Ideas

The first session included generating ideas during an individual brainstorming task, which was sent to the participants via email. The participants first did a “warm-up task” to stimulate understanding of the concept by defining “inadequate sleep” for children aged 4 to 12 years (see Appendix A). Subsequently, participants brainstormed according to the focus statement. Participants were encouraged to fill in as many ideas as possible and to be as specific as possible. Participants returned their completed documents through email. After receiving all documents, the researchers removed duplicate ideas and synthesised the generated ideas into a total set of unique ideas for each group of professionals.

### 2.3. Step (3) Structuring Ideas

The second session included structuring ideas with one sorting task and two rating tasks. The participants completed the tasks using Ariadne; a web service designed specifically for concept mapping. Ariadne allows participants to sort and rate all unique ideas they generated [22]. In the sorting task, participants individually sorted all unique ideas they perceived as fitting together into piles and named each pile. The rules for the sorting process were (1) a minimum of three and a maximum of ten piles have to be created; (2) all ideas need to be sorted into a pile; (3) a pile must consist of a minimum of two ideas; (4) no miscellaneous pile is allowed; and (5) ideas should not be sorted based on their perceived degree of influence on children’s sleep or level of occurrence in practice. In the two rating tasks, the participants rated all ideas on their perceived level of influence on children’s sleep and level of occurrence in practice based on the two rating statements.

### 2.4. Step (4) Analyses

The demographic data were entered into SPSS Statistics 26 [23] and analysed using descriptive statistics. The data from the defining task were analysed using Microsoft Excel [24].

The sorting and rating data were analysed using the online programme Ariadne [22]. Firstly, a group similarity matrix was created to show how often ideas were grouped together by participants. A higher number indicated a greater conceptual similarity between ideas. Secondly, multidimensional scaling of the similarity matrix was conducted. This analysis places each idea as a point on a two-dimensional map. The distance between points represents the conceptual similarity between ideas according to participants. Thirdly, hierarchical cluster analysis was performed. This groups individual ideas into clusters of ideas that reflect similar concepts across participants in each group of professionals. Subsequently, a hierarchical cluster tree was created. This is a schematic figure that presents how the clusters can be optimally arranged based on their conceptual similarity. This figure was used for the interpretation phase.

### 2.5. Step (5) Interpretation

Two researchers (L.S.B. and L.B.) independently selected the final number of clusters using the divisive method with the hierarchical cluster tree [17]. Disagreements were discussed and, if needed, resolved by a third researcher (V.B.). For this method, the description can be found elsewhere [16]. Some ideas were moved to an existing cluster, or a new cluster was created when this made more sense conceptually. For each idea, average rating scores were calculated in Ariadne regarding its influence on children’s sleep and occurrence in practice. For each of the three groups of professionals, a concept map and a go-zone plot were created. A go-zone plot is a bivariate graph in which the average ratings of influence on children’s sleep and occurrence in practice of all ideas are plotted. The final concept maps and go-zone plots can be found in Appendix A, respectively.

Within the three final concept maps, the clusters represented overarching themes of several related perceived determinants, which were defined as categories. To create a final set of perceived determinants across all groups, conceptually similar original ideas were merged into perceived determinants. Subsequently, average ratings for influence on children’s sleep and occurrence in practice were calculated for each perceived determinant, based on the average ratings of the underlying original ideas. Perceived determinants with an average rating ≥3.0 (on a scale of 1–5) for both rating tasks were considered important, i.e., greatly impacting children’s sleep and frequently occurring in practice, as perceived by all three groups of professionals. Ratings between 2.95 and 2.99 were rounded to 2.9. Lastly, all participants were given the opportunity to respond to the results presented in a digital factsheet.

To validate our findings, we conducted a validation study among a new group of professionals of child public health doctors, child public health nurses, parent-and-child advisors, and sleep therapists. Sleep therapists are remedial therapists trained to treat behavioural sleep problems and related physical or mental complaints. They were added to the group for their great practical knowledge of sleep. Through purposive sampling, doctors/nurses were recruited via the Public Health Service of Amsterdam and the Academic Collaborative Centre Child and Youth Health. Sleep therapists were recruited via the Dutch Association of Sleep Therapists. Via email, we asked the participants to assess the conceptual fit of the determinants within the categories (on a five-point Likert scale from “not well at all = 1” to “very well = 5”), to identify any missing perceived determinants and suggest any improvements of the categories’ names. In addition, we asked the participants to assess the importance of the perceived determinants that received an average rating of ≥3.00 in the original study for future healthy sleep interventions (on a five-point Likert scale from “not important = 1” to “very important = 5”). The comprehensiveness and feasibility of the questions in the validation study were pre-tested among four professionals from the Public Health Service of Amsterdam and changes were made accordingly. In the validation study, participants returned their completed documents via email. After receiving all documents, the researchers calculated average ratings for the conceptual fit of each category and the importance of each perceived determinant. Moreover, perceived determinants were moved to a different category and categories were named differently if that made more sense conceptually. Identified missing perceived determinants were included in the final set of perceived determinants if they were not mentioned in the original study. See Appendix A for the results of this validation study.

## 3. Results

### 3.1. Participants

A total of 27 professionals (96% female; 44.7 ± 11.3 years old) participated in the original study and were included in the analysis: nine doctors, eleven nurses, and seven sleep experts. The number of participants differed per task, ranging from 25 to 27 participants. In addition, 16 other professionals (94% female; 43.7 ± 9.9 years old) participated in the validation study: three public health doctors, three public health nurses, one pedagogical advisor, and nine sleep therapists. Table 1 presents the characteristics of the two study samples.

### 3.2. Concept Maps

Professionals generated between 54 and 62 unique ideas per group clustered in four to five categories. All ideas were combined into a final set of 62 perceived determinants (number of unique ideas per group: doctors 52, nurses 62, sleep experts 52), clustered into five categories: psychosocial determinants, daytime and evening activities, medical determinants, pedagogical determinants, and sleep-environmental determinants. The category “daytime and evening activities” was originally only mentioned by nurses. This category includes ideas from the clusters “pedagogical determinants” according to doctors, “daytime and evening activities” according to nurses, and “sleep hygiene” according to sleep experts. In the validation study, the researcher referred to this combined cluster as “behavioural determinants”. Table 2 shows the categories and the perceived determinants of the three groups of professionals combined. The underlying ideas per perceived determinant, per group of professionals can be found in Appendix A. The three final concept maps can be found in Appendix A.

In the validation study, the average conceptual fit rating per category (i.e., how well the determinants fit within the categories), ranged from 3.8 to 4.3. Eleven new perceived determinants were identified and added to the total set of determinants (see Table 2). The name of the category “behavioural determinants” was changed to “daytime and evening activities”, as suggested by four professionals. The results of the validation can be found in Appendix A.

### 3.3. Perceived Determinants of Children’s Inadequate Sleep

The perceived determinants that were rated as important by all groups of professionals were a change in daily life, worrying, using screens before bedtime, playing stimulating games before bedtime, inadequate amount of physical activity at daytime, an inconsistent sleep schedule, and having no bedtime routine.

The three perceived determinants that were mentioned by all groups of professionals with the highest mean rating for influence on children’s sleep and occurrence in practice were (1) using screens before bedtime, (2) playing stimulating games before bedtime, and (3) having an inconsistent sleep schedule.

Some perceived determinants were rated as important by some groups of professionals but were not mentioned by another group. For instance, perceived determinants mentioned by nurses and sleep experts but not doctors were excessive, daytime screen use; inadequate time to relax; the availability of screens in the bedroom; and a disturbed biological clock. A perceived determinant mentioned by doctors and nurses but not sleep experts was parents being unable to provide their children with enough structure during the day. Doctors were the only professional group that mentioned parents’ inability to set boundaries; and only sleep experts mentioned the lack of clear rules. Furthermore, none of the perceived determinants in the sleep-environment category received a mean rating ≥3.00 for both influence on children’s sleep and occurrence in practice.

The three go-zone plots show the importance of the underlying ideas per group of professionals (see Appendix A). In these plots, ideas with a mean rating ≥3.0 for both influence on children’s sleep and occurrence in practice are located in the upper-right quadrant of the plot; the “go-zone”. Ideas that reached the go-zone based on data from all groups of professionals were underlying ideas of the perceived determinants: using screens before bedtime, an inconsistent sleep schedule, having no bedtime routine, worrying, and a change in daily life. For the doctors, most ideas that reached the go-zone were pedagogical factors, while for the nurses most were psychosocial factors, and for the sleep experts most were sleep hygiene factors.

The perceived determinants that were rated as important in the original study were all considered important (mean rating ≥3.0 on a scale of 1–5) for future healthy sleep interventions in the validation study: a change in daily life (3.8), worrying (4.4), using screens before bedtime (4.4), playing stimulating games before bedtime (4.5), inadequate amount of daytime physical activity (4.4), having an inconsistent sleep schedule (4.6), and the lack of a bedtime routine (4.8).

The validation study identified ten additional perceived determinants: negative self-image, parents’ resilience, lack of attention from parent(s), high sensitivity, no time to put child to bed, parents set the wrong example, social norm for playing outside, parents’ ability to respond to the child’s sleep problems, parents’ attitude towards sleep, and absence of stuffed toy.

## 4. Discussion

The aim of this study was to explore the perspectives of professionals with expertise in children’s sleep health on potential determinants of inadequate sleep among children aged 4–12 years. Professionals identified a large variety of potential determinants of children’s inadequate sleep. These determinants were clustered in five categories, i.e., psychosocial determinants, daytime and evening activities, pedagogical determinants, medical determinants, and sleep-environmental determinants. The determinants perceived as most important by all professionals were worrying, a change in daily life, using screens before bedtime, playing stimulating games before bedtime, an inadequate amount of daytime physical activity, an inconsistent sleep schedule, and the lack of a bedtime routine.

Screen use (e.g., television, tablet, phone, or computer) before bedtime was rated as the most important determinant. It is generally acknowledged that agitating screen use before bedtime disturbs children’s sleep [14,15,25]. Underlying mechanisms may include the displacement of sleep time, light exposure, and psychological agitation due to inappropriate content [13,15,25]. Interestingly, while the professionals in this study rated screen use as most important, children themselves rated screen behaviour as least important in an earlier study [16]. Children’s positive experiences with screen use may explain this. Children may not be aware of the potential negative impact pre-bedtime screen use has on their sleep health. For example, a recent study by Cernikova et al. found that children reported sleep problems and tiredness after using digital media. However, they were unaware that this may have been a result of their media use [26]. Furthermore, there are indications that many parents endorse the belief that watching television helps their child fall asleep [27]. This parental attitude towards screen behaviour seems to be related to their children’s screen use [28]. There is corroborating longitudinal evidence for a relationship between screen time (i.e., watching TV, using a computer or gaming) and children’s sleep health [14]. However, the agreement between this scientific evidence and professionals’ perception may also be a result of professionals’ knowledge of this evidence. Furthermore, there is insufficient evidence on how much and what types of evening screen use are most disruptive to children’s sleep health [27]. There is some evidence that interactive screen time (e.g., mobile phone or video games) is more agitating than passive use (e.g., watching TV) and that the nature of the content is important (i.e., violent content is more agitating) [27]. Evidently, these are interesting topics for future research. One recent meta-analysis showed that interventions that control screen use can have a positive impact on children’s sleep [29]. It therefore seems relevant to include limitation of screen time in future healthy sleep interventions, especially screen time before bedtime.

Another important difference between the perspectives of professionals, parents, and children is related to medical- and sleep-environmental determinants. Although several medical and sleep-environmental determinants were mentioned by professionals, they rated none as important. Compared to children and parents, professionals rated medical determinants as relatively less important [16]. This may be because the underlying ideas were different; professionals mentioned diseases and disorders, while children and parents referred to having a cold or having pain in general. Moreover, children rated some parts of their sleep environment (e.g., comfort of their bed and the temperature in their bedroom) as one of the most important potential determinants, while professionals and parents considered children’s sleep environment less important [16]. These findings highlight that various involved stakeholders can have a different perspective on the importance of potential determinants of children’s inadequate sleep health. Therefore, professionals working in child public health are recommended to consider children’s and parents’ perspectives in their advice and treatment regarding children’s sleep health.

Our results underpin the crucial role of parents in their children’s sleep health. Many perceived determinants identified in the categories psychosocial, sleep environment, daytime and evening activities, involve parents. For example, the perceived determinants “an upcoming stressful event” and “unprocessed thoughts or feelings” are less stressful for children when their parents help them cope with their feelings and emotions. This confirms that parenting behaviour is strongly related to children’s sleep hygiene at this age. To illustrate this, children’s screen use depends on the rules set by their parents, and children are more likely to be physically active during the day when this behaviour is encouraged by their parents [30,31]. Teaching parents skills to effectively limit their child’s agitating screen use before and after bedtime and encourage physical activity during the day may contribute to positive changes in their child’s sleep health [15,32,33]. Hence, it seems valuable for future interventions to focus on improving parenting skills in order to change children’s behaviour and, subsequently, their health [34].

Many of the perceived determinants identified by professionals in this study were also identified by children and parents [16]. As this study explored the perspectives of professionals on potential determinants, the results of this study do not provide evidence for a relationship between the identified factors and children’s sleep health. For many of the identified factors, there is at least some evidence for an association with child sleep health [14,15]. However, further longitudinal research is required to examine whether the identified factors are actual determinants of children’s sleep.

From the wide range of identified potential determinants in several determinant domains, we can conclude that children’s inadequate sleep health is a multi-factorial problem. Furthermore, many of the determinants may be interrelated. Finally, every determinant has several stakeholders at different levels of the social ecological model [35,36]. For example, the determinant “structural daytime and evening routines” includes parents and other family members (e.g., siblings, grandparents) as stakeholders, while the determinant “noise” includes other stakeholders such as neighbours and city planners. These stakeholders may also be interrelated. This makes children’s inadequate sleep not only a multi-factorial but also a complex problem. A complex problem can be defined as a problem that is difficult to demarcate and define, with no fixed amount of possible solutions, and where everything is interrelated and dynamic [37]. This complexity means a single solution will be insufficient to structurally promote healthy sleep for all children, because determinants and, in effect, adequate solutions may vary widely. One way to tackle this problem could be to use “systems thinking” for the development of preventive healthy sleep interventions for children. Systems thinking means capturing the problem as a whole rather than its individual parts, while taking into account the interrelatedness between these parts and the dynamic character of the health behaviour, in this case sleep behaviour, to make sense of a complex situation [38,39].

The identified determinants and the clustering of these determinants differed somewhat between the three groups of professionals. One reason may be that the online process included an individual brainstorm session rather than a group brainstorm, which could have yielded new ideas more easily. An idea for future online concept mapping sessions is to organise a collective online brainstorm session. The differences between the three groups of professionals may also be ascribed to the nature of their profession. Whereas the groups consisting of child public health nurses and doctors were more generally educated on children’s health and saw children with a wide variety of problems, the group of sleep experts included professionals who specialised in children’s sleep health and mostly saw children with sleep-related problems or performed research in this field. Furthermore, only the group of sleep experts identified a cluster “sleep hygiene”. This cluster contained several types of determinants, such as psychosocial, pedagogical, and sleep-environmental, of which many were similar to ideas in the cluster “daytime and evening activities” identified by the group of nurses. An explanation for this different clustering between groups may be a difference in the terminology used in different fields of expertise.

According to the professionals participating in this study, future healthy sleep interventions should at least promote a regular structure of daytime and evening routines; child-appropriate, set bedtimes; an adequate amount of relaxation before bedtime; and the avoidance of agitating activities that can increase alertness or anxiety. These determinants were identified as go-zones by all groups of professionals. The go-zones identified by doctors were mainly pedagogical-environmental determinants, while those identified by nurses were mostly psychosocial. Most of these go-zones were clustered as sleep hygiene determinants by the sleep experts. The go-zone plot of sleep experts showed more than half of the generated ideas in the “go-zone”, which means that they rated these ideas as both important and often occurring in practice. This corresponds to the occupation they perform. Little is known about the effectiveness of these appointed “go-zones”. One previous review [40] found that denominators across effective interventions included focussing on multiple health behaviours and intervention settings and integrating multiple behaviour change strategies that target the most important determinants. There is a need for more high-quality evaluation studies on interventions that promote children’s sleep health.

The validation study confirmed the adequacy of the original clustering and importance rating. The validation sample also identified some missing determinants: negative self-image, parents’ resilience, lack of attention from parent(s), high sensitivity, no time to put child to bed, parents set the wrong example, social norm for playing outside, parents’ ability to respond to the child’s sleep problems, parents’ attitude towards sleep, and absence of stuffed toy. This may be explained by the specific expertise of the additional group of professionals in the validation study: sleep therapists specialised in children’s sleep problems. Another reason may be the lack of a group brainstorm session in the original study.

### Strengths and Limitations

A strength of this study is that it provides additional insights into potential determinants of children’s inadequate sleep health, which are crucial for the development of future interventions promoting healthy sleep among children. Another strength is that the group of participants was diverse in terms of occupation and included the most prominent Dutch health professionals who deal with child sleep health. Sleep experts are often involved in research into children’s sleep, while child public health doctors and nurses are in close contact with children themselves. This provided perspectives from both research and practice. Additionally, the use of the concept mapping method allowed us to collect a broad range of unique ideas on all aspects of inadequate sleep from various perspectives. The validation of our results in the validation study among a second group of professionals further strengthens our study.

However, this study is not without limitations. Firstly, the concept mapping sessions were held online, which did not allow for a group brainstorm as part of idea generation. This may have led to potential misinterpretations of ideas that were presented to the professionals in the sorting and rating task. Secondly, this study focusses on the age range of 4 to 12 years old. Since this is a relatively wide age range, with varying behaviours, needs, and parenting strategies, not all identified determinants may be relevant for every age within this range. Lastly, the use of a focus statement requires a focus on either adequate or inadequate sleep health. This led to omitting potential determinants of adequate sleep health.

## 5. Conclusions

This study identified determinants of children’s inadequate sleep health as perceived by professionals with expertise in children’s sleep health in five categories, i.e., psychosocial determinants, daytime and evening activities, pedagogical determinants, medical determinants, and sleep-environmental determinants. From the wide range of identified potential determinants, we can conclude that children’s inadequate sleep is multi-factorial. Therefore, developing an intervention to promote children’s healthy sleep may benefit from a systems approach, including all relevant factors, stakeholders, and their interrelations.

## Figures and Tables

**Figure 1 ijerph-17-07315-f001:**
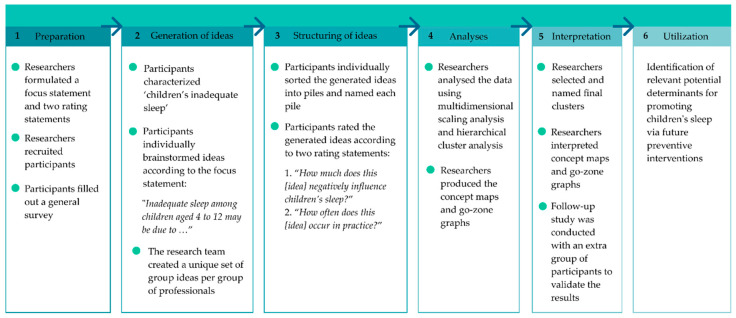
The six-step concept mapping process [16,17,18].

**Table 1 ijerph-17-07315-t001:** Study sample characteristics.

Characteristics	Original Study Sample			Validation Study Sample
	Doctors (*N* = 9)	Nurses (*N* = 11)	Sleep Experts (*N* = 7)	Mixed Group of Professionals (*N* = 16)
Age (years, *M, SD*)	40.3 (8.3)	48.1 (13.0)	45.1 (11.6)	43.7 (9.9)
Female (%)	88.9	100.0	100.0	93.8
Working experience (years, *M, SD*)	10.6 (6.3)	13.3 (10.8)	15.7 (12.1)	5.3 (3.6)
Share of sleep in their work (%)				
Large	33.3	18.2	100.0	50.0
Moderate	55.6	54.5	0.0	37.5
Small	11.1	27.3	0.0	12.5

*N* = number of participants; M = mean; SD = standard deviation.

**Table 2 ijerph-17-07315-t002:** Mean rating of influence on sleep ^1^ and occurrence in practice ^2^ for the professional-perceived determinants of children’s inadequate sleep.

Category	Perceived Determinants	Examples of Underlying Original Ideas	Mean Rating per Professionals’ Group						Mean Rating ^3^	
			Doctors		Nurses		Sleep Experts			
			In	Oc	In	Oc	In	Oc	In	Oc
**Psychosocial determinants**									*3.7*	*2.7*
	Being bullied	“Being bullied and worrying about this in the evening/at night”	N.A.	N.A.	4.3	2.8	4.3	2.7	4.3	2.8
	A change in daily life	“Changes in the home situation: divorce, death of a loved one”, “Changes in daily life: moving, new school, holidays”	3.6	3.2	4.0	3.0	4.3	3.0	**4.0**	**3.1**
	Worrying	“Worrying about the day that has been”, “Worrying in the evening/at night about problems at school”	3.7	2.9	4.0	2.8	4.0	3.4	**3.9**	**3.0**
	Stressful family situation	“Experiencing stress due to arguing at home”	4.3	2.5	4.3	2.9	4.0	2.7	4.2	2.7
	Traumatic event	“A traumatic experience from the past”	4.1	2.2	4.3	2.7	3.9	2.6	4.1	2.5
	Upcoming stressful event	“Feeling stressed about something that will happen the next day”	N.A.	N.A.	3.6	2.8	N.A.	N.A.	3.6	2.8
	Unprocessed thoughts or feelings	“Having to process the day”	3.3	2.8	3.5	3.0	N.A.	N.A.	3.4	2.9
	Bedtime resistance ^4^	“Refusing to go to bed”	N.A.	N.A.	3.3	2.6	3.9	2.9	3.6	2.7
	Nightmares	“Scary or bad dreams that wake a child”	3.6	2.8	3.6	2.6	3.6	2.7	3.6	2.7
	Performance pressure	“Feeling pressure to perform well”	N.A.	N.A.	3.5	2.8	N.A.	N.A.	3.5	2.8
	Feeling unsafe	“Not feeling safe at home or at school”	3.8	2.1	4.2	2.3	4.0	2.3	4.0	2.2
	Fear	“Scared of parent(s) leaving”, “Fear of the dark”	3.5	2.2	3.8	2.9	3.7	2.3	3.6	2.5
	Loneliness	“Missing a parent or waiting for a parent to come home”	N.A.	N.A.	3.5	2.5	N.A.	N.A.	3.5	2.5
	Parental stress	“Parental stress and agitation that is transferred to the child”	N.A.	N.A.	N.A.	N.A.	3.3	2.6	3.3	2.6
	Child’s temperament	“The child’s temperament”	N.A.	N.A.	2.6	3.1	N.A.	N.A.	2.6	3.1
	Negative self-image ^5^	“Negative self-image”	-	-	-	-	-	-	-	-
	Parents’ resilience ^5^	“Parents’ support system”, “Parents’ mental health”	-	-	-	-	-	-	-	-
	Lack of attention from parent(s) ^5^	“Need for undivided attention from parents”	-	-	-	-	-	-	-	-
**Daytime and evening activities ^6^**									*3.3*	*3.0*
	Screens in the bedroom	“Free use of screens (TV, phone, tablet, computer) in the bedroom”	N.A.	N.A.	4.5	3.4	N.A.	N.A.	**4.5**	**3.4**
	Screen use before bedtime	“Screen use (TV, phone, tablet, computer) right before sleeping”, “Watching a movie until late before going to sleep”	4.1	4.0	3.9	3.4	4.1	3.9	**4.0**	**3.7**
	Playing stimulating games before bedtime	“Gaming/Playing on the game console until right before going to sleep” or “Too busy with playing around just before going to bed”	3.9	3.8	3.4	2.6	4.4	3.7	**3.9**	**3.4**
	Excessive daytime screen use	“Excessive use of computer or other screens (tablet, phone) during the day”, “Excessive gaming/playing on the game console during the day”	N.A.	N.A.	3.4	3.8	2.9	3.6	**3.1**	**3.7**
	Inadequate time to relax	“Not relaxing enough before going to bed, causing the brain to be too active”, “Doing homework until late in the evening”	N.A.	N.A.	3.3	3.5	3.9	3.4	**3.6**	**3.5**
	Inadequate amount of daytime PA	“A lack of exercise during the day”	2.9	3.2	2.9	3.1	3.6	3.4	**3.1**	**3.2**
	Sleep-disrupting food or drinks	“Drinking stimulating (sugary and/or caffeinated) drinks in the evening”	3.8	2.4	N.A.	N.A.	3.4	3.0	3.6	2.7
	Too much evening light	“Too much light in the evening”	N.A.	N.A.	N.A.	N.A.	3.4	2.9	3.4	2.9
	Bed also used for other activities	“Using the bed for more than just sleeping: playing in bed, watching TV in bed”	N.A.	N.A.	N.A.	N.A.	2.7	3.4	2.7	3.4
	Inadequate time outside at daytime	“Too little light and fresh air during the day due to being inside too much”	2.9	3.1	2.6	2.8	3.1	3.3	2.9	3.1
	Inadequate morning light	“Too little light in the morning”	N.A.	N.A.	N.A.	N.A.	3.1	2.7	3.1	2.7
	Evening PA	“Being physically active/doing sports late in the evening”	3.2	2.0	2.6	2.2	3.9	3.0	3.2	2.4
	Too many daytime activities	“Too many daytime activities”	N.A.	N.A.	3.0	2.4	N.A.	N.A.	3.0	2.4
	Eating/drinking close to bedtime	“Eating/drinking too late”	2.8	2.3	2.3	2.4	3.0	2.4	2.7	2.4
**Medical determinants**									*3.6*	*2.5*
	Disturbed biological clock	“A disturbed biological clock”	N.A.	N.A.	4.0	2.5	4.4	3.9	**4.2**	**3.2**
	Illness	“Illness/pain”	3.1	3.0	3.7	2.1	4.3	2.9	3.7	2.7
	Mental problems	“Mental condition”	3.3	2.2	3.6	2.4	4.0	3.4	3.7	2.7
	Medication use with effects on sleep	“Using melatonin (sleeping pills) wrongly”	3.6	2.1	4.0	2.0	3.9	3.1	3.8	2.4
	Sleep disorder	“Wetting the bed”, “Sleep walking”	3.9	2.3	2.6	2.0	4.5	3.4	3.7	2.6
	Physical complaint	“Cold feet”, “Growing pain in the legs	4.3	2.6	N.A.	N.A.	3.6	1.9	4.0	2.2
	Medical complaint	“Overweight/obesity”, “Allergy/itch	3.0	2.7	2.8	2.6	3.9	2.2	3.2	2.5
	Heredity for sleep problems	“Heredity for sleep problems/hereditary”	2.8	1.8	N.A.	N.A.	N.A.	N.A.	2.8	1.8
	High sensitivity ^5^	“High sensitivity”	-	-	-	-	-	-	-	-
**Pedagogical determinants**									*3.2*	*2.8*
	No consistent sleep schedule	“Having irregular sleep times”, “Going to bed too late/Being put to bed too late”	3.7	3.3	3.4	3.7	4.0	3.6	**3.7**	**3.5**
	Inadequate structure of the day	“Having no rhythm and little structure in the daily routine”	3.9	3.11	3.5	3.4	N.A.	N.A.	**3.7**	**3.3**
	Parents’ inability to set boundaries	“Parents who find it difficult to set limits”	3.4	3.3	N.A.	N.A.	N.A.	N.A.	**3.4**	**3.3**
	No bedtime routine	“Having no set sleep/bedtime routine”	3.4	3.3	3.3	3.1	3.6	3.3	**3.4**	**3.2**
	Lack of clear rules	“Parents who do not set clear rules or are too strict”	N.A.	N.A.	N.A.	N.A.	3.3	3.3	**3.3**	**3.3**
	Daytime nap	“Sleeping (too much) at daytime”	3.4	2.4	3.9	2.4	4.0	2.3	3.8	2.4
	Inadequate bedtime ^7^	“Going to bed too early or being put to bed too early”	2.9	1.9	N.A.	N.A.	3.9	3.3	3.4	2.6
	Parents’ negative focus on sleep	“Parents who create negative attention around sleep by putting too much emphasis on having to sleep”	N.A.	N.A.	3.4	2.5	3.4	2.6	3.4	2.5
	Parents each applying different bedtime rules	“(Separated) parents applying different rules regarding bedtimes”	N.A.	N.A.	N.A.	N.A.	3.3	2.4	3.3	2.4
	Co-sleeping	“Sleeping in bed with parents or other family member”	3.0	2.8	2.1	2.8	3.0	3.4	2.7	3.0
	Bedtime procrastination	“Children demand a too elaborate bedtime ritual before going to sleep”	3.1	2.6	2.4	2.2	3.7	3.1	3.1	2.6
	Insufficient parental sleep-related knowledge	“Uncertainty among parents about what an appropriate bedtime is”	N.A.	N.A.	2.1	2.8	N.A.	N.A.	2.1	2.8
	Social norm for bedtime at school	“Comparing oneself to classmates who go to bed later”	N.A.	N.A.	2.2	2.7	N.A.	N.A.	2.2	2.7
	Inadequate amount of food or drink	“Being hungry or thirsty”	N.A.	N.A.	2.7	2.0	N.A.	N.A.	2.7	2.0
	No time to put child to bed ^5^	“Parents do not put their child to bed on time because they prioritise other matters, e.g., work, being home late, cleaning”	-	-	-	-	-	-	-	-
	Parents set the wrong example ^5^	“Setting the wrong example regarding screen use”	-	-	-	-	-	-	-	-
	Social norm for playing outside ^5^	“The norms for playing outside”	-	-	-	-	-	-	-	-
	Parents’ ability to respond to the child’s sleep problems ^5^	“Knowledge, ability, and skills to react adequately to night terrors, trouble falling asleep, or sleeping through the night”	-	-	-	-	-	-	-	-
	Parents’ attitude towards sleep ^5^	“Parents’ thoughts on sleep”	-	-	-	-	-	-	-	-
**Sleep-environmental determinants**									*2.8*	*2.1*
	No bedroom	“Not having a bedroom or sleeping in another place in the house”	3.1	2.7	N.A.	N.A.	N.A.	N.A.	3.1	2.7
	Bedroom sharing	“Sleeping with multiple people in a room”	2.7	3.1	2.7	3.0	3.7	2.0	3.1	2.7
	Noise outside	“Noise in the neighbourhood/outside”	3.1	2.1	3.1	2.5	3.0	1.9	3.1	2.2
	Cramped housing	“Living in a small house with a lot of people”	N.A.	N.A.	2.6	2.5	N.A.	N.A.	2.6	2.5
	Noise inside	“Noise in the house”	N.A.	N.A.	2.7	2.3	3.0	2.0	2.9	2.2
	Uncomfortable bed	“Uncomfortable bed”	N.A.	N.A.	N.A.	N.A.	3.0	2.0	3.0	2.0
	Too much light	“Not dark enough in the bedroom”	2.6	1.8	2.7	1.6	3.0	2.4	2.8	2.0
	Not the right temperature	“Too cold or too warm in the bedroom”	3.0	2.0	2.8	1.7	2.7	1.9	2.8	1.9
	Not ventilated	“Room not ventilated”	2.7	1.9	N.A.	N.A.	N.A.	N.A.	2.7	1.9
	Untidy bedroom	“Too much clutter or stuffed pets in the room”	N.A.	N.A.	1.9	1.4	2.9	2.3	2.4	1.9
	Uncomfortable sleep wear	“Uncomfortable sleep wear”	N.A.	N.A.	N.A.	N.A.	2.6	1.1	2.6	1.1
	Absence of stuffed toy ^5^	“Safety through stuffed toy”	-	-	-	-	-	-	-	-

In = influence on sleep; Oc = occurrence in practice; N.A. = not applicable, i.e., this item was not mentioned in this group of professionals; PA = physical activity. ^1^ The average influence on sleep rating is based on the question: “How much does this [idea] negatively influence sleep?” answered on a five-point Likert scale from “no influence at all = 1” to “great influence = 5”. ^2^ The average occurrence in practice rating scale based on the question: “How often does this [idea] occur in practice?” answered on a five-point Likert scale from “never = 1” to “always = 5”. ^3^ Mean = mean rating for influence on children’s sleep or occurrence in practice across all groups of professionals. ^4^ This perceived determinant was moved from category “Pedagogical determinants” to category “Psychosocial determinants” according to the results of the validation study. ^5^ This perceived determinant was added according to the results of the validation study and was therefore not rated by the original group of participants. ^6^ This name was changed from “Behavioural determinants” to “Daytime and evening activities” according to the results of the validation study. ^7^ This perceived determinant was moved from category “Daytime and evening activities” to category “Pedagogical determinants”. Bold values indicate that the category or determinant was perceived as important (i.e., influence on sleep and occurrence in practice ratings ≥3.00); Values in italic indicate the overall mean rating per cluster.

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
