# Peer review of "Perceived Determinants of Children’s Inadequate Sleep Health. A Concept Mapping Study among Professionals"

_ijerph, 2020, doi:10.3390/ijerph17197315_

Round 1

Reviewer 1 Report

The authors have tried to investigate potential determinants of children’s inadequate sleep health through a questionnaire to 3 different group of experts. The study is interesting and, although at a first glance it seems very complicated, the supplementary material and the tables are very helpful. The use of Ariadne was a quite clever tool and helpful in order to underline differences of the influences on children’s sleep and occurrence in practice and to highlight the differences among the three groups of professionals in the identified determinants and the clustering of these determinants, a fact that per se shows the complexity of the issue. 

The authors recognize that the problem is multifoctorial the thus the need for further research and more high-quality evaluation studies is imperative in order to suggest the necessary interventions that promote children’s sleep health they point out They also admit that there are limitations. i.e the age range of 4 to 12 years old includes children of different stages of puberty, from prepuberty to almost end of puberty, a fact that possibly changes the habits of the children and consequently the determinants of inadequate sleep health; in addition, the sex should also be added in a future study since boys and girls behave differently.

Comment

Was sex of children  considered as a different parameter by the 3 groups of participants?? If not it  should propably be added in the Limitations along with the age groups.  

English minor editing errors  

Supplementary Materials: The following online at www.mdpi.com/xxx/s1, Figure S1: title, Table 398 S1: title, Video S1: title was not found 

Author Response

We thank the reviewer for his/her time to review our manuscript and the valuable feedback. Below we provide a point-to-point reply to the reviewer’s comments.

Comment 1.1

The authors have tried to investigate potential determinants of children’s inadequate sleep health through a questionnaire to 3 different group of experts. The study is interesting and, although at a first glance it seems very complicated, the supplementary material and the tables are very helpful. The use of Ariadne was a quite clever tool and helpful in order to underline differences of the influences on children’s sleep and occurrence in practice and to highlight the differences among the three groups of professionals in the identified determinants and the clustering of these determinants, a fact that per se shows the complexity of the issue.

Response 1.1

We thank the reviewer for the compliments on our study.

Comment 1.2

The authors recognize that the problem is multifactorial the thus the need for further research and more high-quality evaluation studies is imperative in order to suggest the necessary interventions that promote children’s sleep health they point out They also admit that there are limitations. i.e. the age range of 4 to 12 years old includes children of different stages of puberty, from prepuberty to almost end of puberty, a fact that possibly changes the habits of the children and consequently the determinants of inadequate sleep health; in addition, the sex should also be added in a future study since boys and girls behave differently. Was sex of children considered as a different parameter by the 3 groups of participants?? If not it  should probably be added in the Limitations along with the age groups. 

Response 1.2

We thank the reviewer for this suggestion. We agree that boys and girls behave differently and that future studies could explore gender-specific determinants of sleep behavior. However, our study focused on potential determinants of sleep behavior in general from the perspective of professionals and not on gender differences. Therefore, gender differences were not included in the design of our study.

Comment 1.3

English minor editing errors.

Response 1.3

We thank the reviewer for noting a minor spell check. Therefore, we have checked our manuscript on errors. The full manuscript was also checked by an English language editing service before submission.

Reviewer 2 Report

Thank you for possibility of reviewing this interesting manuscript. Your article has many merits and has been prepared carefully. This research revealed many important clues that may help both parents and specialists to have a better understanding of children's inadequade sleep health.

The strongest points of the article are:

  • readability, logical argument and good selection of sources which are a good rationale for conducting this research
  • pracitical implications which have been clearly stated and directions of further research in Discussion section
  • interesting method of gathering data (and its detailed description step by step)

Below I list some suggestions which can be made to improve the article's quality:

  • why Author/Authors decided to take into consideration children between 4 and 12 years? In other words, what is the reason of analysis of the quality and quantity of children's sleep health between 4 and 12? Is it a researcher's scientific own idea/design or maybe there is something typical in the the way how children at this age range sleep and therefore, there is a reason of putting them together in one cohort and ask specialists about potential underpinnings of their problems with sleep? It would be recommendable to put a short explanation in Introduction section
  • Table 1 - MD, SD should be written in italics
  • it is recommended to unify parantheses that are used e.g. in lines 82 "no influence at all = 1" and 174: 'not important = 1'

Author Response

We thank the reviewer for his/her time to review our manuscript and the valuable feedback. Below we provide a point-to-point reply to the reviewer’s comments.

Comment 2.1

Thank you for possibility of reviewing this interesting manuscript. Your article has many merits and has been prepared carefully. This research revealed many important clues that may help both parents and specialists to have a better understanding of children's inadequate sleep health.

The strongest points of the article are:

  • readability, logical argument and good selection of sources which are a good rationale for conducting this research
  • practical implications which have been clearly stated and directions of further research in Discussion section
  • interesting method of gathering data (and its detailed description step by step)

Response 2.1

We thank the reviewer for the compliments on both our study and article.

Comment 2.2

Why Author/Authors decided to take into consideration children between 4 and 12 years? In other words, what is the reason of analysis of the quality and quantity of children's sleep health between 4 and 12? Is it a researcher's scientific own idea/design or maybe there is something typical in the way how children at this age range sleep and therefore, there is a reason of putting them together in one cohort and ask specialists about potential underpinnings of their problems with sleep? It would be recommendable to put a short explanation in Introduction section.

Response 2.2

We agree with the reviewer that this was unclear in our manuscript and added this information to the beginning of the methods section. We acknowledge that because of the relatively wide age range of 4 to 12 years old, the identified perceived determinants may differ for different age groups. This is included in the Discussion section of the manuscript.

This now reads…
  • “An online concept mapping study was conducted to identify professionals’ perspectives on potential determinants of inadequate sleep health among children aged 4-12 years [17]. This age group was selected as this marks the period for primary school in the Netherlands.” [Methods, line 65-68]
  • “Secondly, this study focusses on the age range of 4 to 12 years old. Since this is a relatively wide age range, with varying behaviours, needs, and parenting strategies, not all identified determinants may be relevant for every age within this range.” [Discussion, line 394-396]

Comment 2.3

Table 1 - MD, SD should be written in italics.

Response 2.3

We made the changes in Table 1 as suggested by the reviewer.

Comment 2.4

It is recommended to unify parentheses that are used e.g. in lines 82 "no influence at all = 1" and 174: 'not important = 1'.

Response 2.4

Thank you for noticing this. We made the changes accordingly and unified the parentheses throughout the manuscript.

Reviewer 3 Report

This is a very suitable study with an interesting method of data collection. This can be the reason for the big extension of the chapter "material and methods". Even so, I would advise the authors to consider whether they can shorten it by a few sentences, provided that the interesting information is not lost.

As for the table 2, it contains so much information that it is similar to reading a text. I would eliminate the “examples”. They are available in the supplementary material.

Finally, I would add in the weak points of the work, that its results are based on personal opinions of experts, in one way or another, in the field. Despite having used a very careful method, the basis of the analysis is opinions and not facts demonstrated by evidence. This fact does not detract from the value of the study, but it is important to make clear to the reader

Author Response

We thank the reviewer for his/her time to review our manuscript and the valuable feedback. Below we provide a point-to-point reply to the reviewer’s comments.

Comment 3.1

This is a very suitable study with an interesting method of data collection. This can be the reason for the big extension of the chapter "material and methods". Even so, I would advise the authors to consider whether they can shorten it by a few sentences, provided that the interesting information is not lost.

Response 3.1

We thank the reviewer for the compliments on the method that we used. We agree with the reviewer that the methods section could be more concise and therefore shortened it as suggested.

Comment 3.2

As for the table 2, it contains so much information that it is similar to reading a text. I would eliminate the “examples”. They are available in the supplementary material.

Response 3.2

We thank the reviewer for this suggestion. However, we prefer to present the examples in Table 2 because they are the specific statements given by the professionals, while the determinant names were chosen by the researchers. Furthermore, the examples clarify the perceived determinants without it being necessary to check the appendices.

Comment 3.3

Finally, I would add in the weak points of the work, that its results are based on personal opinions of experts, in one way or another, in the field. Despite having used a very careful method, the basis of the analysis is opinions and not facts demonstrated by evidence. This fact does not detract from the value of the study, but it is important to make clear to the reader

Response 3.3

We thank the reviewer for raising this point. The aim of our study was to explore the perspectives of professionals on potential determinants of children’s sleep health. As our aim was to gain insight in such personal opinions we did not mention it as a limitation. However, we agree that we need to emphasize this focus on perceived determinants and have therefore changed the title and discussion section as follows:

  • Perceived determinants of children’s inadequate sleep health. A concept mapping study among professionals.” [Title, line 1-4]
  • “These perspectives are valuable for future longitudinal studies on determinants of children’s sleep and the development of future healthy sleep interventions.” [Abstract, line 30-32]
  • “Many of the perceived determinants identified by professionals in this study were also identified by children and parents [16]. As this study explored the perspectives of professionals on potential determinants, the results of this study do not provide evidence for a relationship between the identified factors and children’s sleep health. For many of the identified factors, there is at least some evidence for an association with child sleep health [14, 15]. However, further longitudinal research is required to examine whether the identified factors are actual determinants of  children’s sleep.”  [Discussion, line 324 – 330]
